# Perceptual formant discrimination during speech movement planning

**Hantao Wang**, **Yusuf Ali**, **Ludo Max***

Department of Speech and Hearing Sciences, University of Washington, Seattle, Washington, United States of America

* LudoMax@uw.edu

**Data Availability Statement:** Data files are available from the Open Science Framework at https://osf.io/sf29z.

**Funding:** This research was supported by grants R01 DC017444 and R01 DC020162 (to author L. M.) and T32 DC005361 from the National Institute

## Abstract

Evoked potential studies have shown that speech planning modulates auditory cortical responses. The phenomenon's functional relevance is unknown. We tested whether, during this time window of cortical auditory modulation, there is an effect on speakers' perceptual sensitivity for vowel formant discrimination. Participants made same/different judgments for pairs of stimuli consisting of a pre-recorded, self-produced vowel and a formant-shifted version of the same production. Stimuli were presented prior to a "go" signal for speaking, prior to passive listening, and during silent reading. The formant discrimination stimulus /uh/ was tested with a congruent productions list (words with /uh/) and an incongruent productions list (words without /uh/). Logistic curves were fitted to participants' responses, and the just-noticeable difference (JND) served as a measure of discrimination sensitivity. We found a statistically significant effect of condition (worst discrimination before speaking) without congruency effect. Post-hoc pairwise comparisons revealed that JND was significantly greater before speaking than during silent reading. Thus, formant discrimination sensitivity was reduced during speech planning regardless of the congruence between discrimination stimulus and predicted acoustic consequences of the planned speech movements. This finding may inform ongoing efforts to determine the functional relevance of the previously reported modulation of auditory processing during speech planning.

## Introduction

Behavioral and neurophysiological studies of both limb and speech movements consistently show that sensory processing is modulated during active movements [1, 2]. Such movement-induced sensory modulation is commonly interpreted in the framework of predictive motor control, in which the sensorimotor system uses the predicted sensory consequences of a motor command to generate and control the movement [3–5].

For example, using extracranial neural recording techniques such as electroencephalography (EEG) and magnetoencephalography (MEG), many studies with human participants have shown attenuated brain responses to either movement-generated sensory feedback [6–9] or external sensory stimuli delivered during movement [6, 10, 11]. On the other hand, intracranial recordings in both humans and animal models have revealed a more complex picture,

on Deafness and Other Communication Disorders (https://www.nidcd.nih.gov/). The content is solely the responsibility of the authors and does not necessarily represent the official views of the National Institute on Deafness and Other Communication Disorders or the National Institutes of Health. The funders had no role in study design, data collection and analysis, decision to publish, or preparation of the manuscript.

**Competing interests:** The authors have declared that no competing interests exist.

with vocalization or limb movements leading to both suppression and facilitation across different neuronal populations or single cells in auditory or somatosensory cortex [12–18]. At the behavioral level, many studies have found that limb movements lead to an attenuated perception of self-generated tactile, auditory or visual stimuli, including increased perceptual thresholds and decreased sensitivity [19–23]. However, several recent studies have shown that movement modulates perception in a complex manner. For example, as compared with externally generated stimuli, the perception of self-generated sensory feedback was found to be enhanced at low stimulus intensity but attenuated at high stimulus intensity [24, 25], or enhanced at an early time point and attenuated at a later time point [26, 27].

In order to fully understand how active movements affect sensory processing and perception, one also needs to consider that the auditory and somatosensory systems are already modulated during vocalization and limb movement planning prior to movement onset [13, 18, 28–32]. Here, we address this topic further in the context of our laboratory's series of speech studies that compared long-latency auditory evoked potentials (AEPs) elicited by pure tone probe stimuli delivered during the planning phase in a delayed-response speaking task versus control conditions without speaking. The probe tone stimulus was always delivered 400 ms after initial presentation of a word and 200 ms prior to the *go* signal cueing over production (or at the equivalent time point in the passive listening and/or silent reading control conditions). Results consistently indicated that the amplitude of the cortical N1 component in the AEPs is reduced prior to speaking [32–35]. However, the functional relevance of this pre-speech auditory modulation (PSAM) phenomenon remains entirely unknown.

To date, only Merrikhi and colleagues [36] have investigated potential perceptual correlates of PSAM. Adapting the delayed-response tasks used in our previous PSAM studies, they asked participants to compare the intensity of two pure tone stimuli in both speaking and no-speaking conditions. The *standard* stimulus with a fixed intensity level was presented at the beginning of each trial. The *comparison* stimulus with varying intensity was presented during the speech planning phase at the time point where PSAM had been previously demonstrated. Based on a two-interval forced choice intensity discrimination test ("Which one was louder?"), the speaking condition showed (a) a statistically significantly higher point of subjective equality (i.e., the comparison stimulus had to be louder to be perceived equally loud as the standard stimulus), and (b) statistically significantly lower slope values for the psychometric functions (i.e., greater uncertainty in the perceptual judgments). Thus, results were consistent with the idea that PSAM during speech movement planning is associated with an attenuation in the perception of auditory input. However, given that [36] tested only intensity perception and only used pure tones, it remains to be determined whether speech planning modulates other perceptual processes with more direct relevance for monitoring auditory feedback once speech is initiated. For example, it is possible that prior to speech onset neuronal populations with different characteristics are already selectively inhibited and facilitated to suppress the processing of irrelevant events but enhance the processing of speech-related auditory feedback. In addition, it is not clear to what extent the results of [36] may have been influenced by a working memory component. In their paradigm, each trial's time interval between the standard and comparison stimuli was 900 ms, and the target word to be spoken, read, or listened to by the participant always appeared on a computer monitor during this comparison interval.

To further investigate the functional relevance of PSAM, the current study combined our prior delayed-response speaking task paradigm with a novel same/different formant discrimination perceptual test that used brief stimuli derived from the participant's own speech and delivered as a pair centered around the exact time point where PSAM has been previously demonstrated [32–35, 37]. The first two formants, or resonance frequencies of the vocal tract, are critical for the production of vowels, and speakers are sensitive to formant frequency

changes perceived in their auditory feedback [38–43]. We therefore asked participants to make the same/different judgments for comparison stimuli that were created by extracting and truncating a pre-recorded self-produced vowel, and digitally altering the formant frequencies of the second stimulus. We controlled the congruency between these stimuli for the formant discrimination test and the vowel of the words in the delayed-response speaking task by using two different word lists: one list only included words containing the same vowel as the discrimination stimuli and the other list excluded words containing that vowel. Silent reading (Reading condition) and passive listening (Listening condition) were included as control conditions. We hypothesized that, if auditory feedback monitoring starts being suppressed during the speech planning phase, formant discrimination sensitivity would decrease in the Speaking condition as compared with the control conditions. Alternatively, if the auditory system is selectively tuned to the predicted acoustic outcomes of the planned speech movements, formant discrimination in the Speaking condition may be enhanced, especially when the discrimination stimuli are congruent with the vowel in the predicted acoustic outcome (i.e., planned production).

## Materials and methods

### Participants

Twenty-six right-handed adult native speakers of American English (16 women, 10 men, age $M = 22.90$ years, $SD = 4.68$ years, range = 18–36 years) with no self-reported history of speech, hearing, or neurological disorders participated between 07/07/2021 and 07/08/2022. Based on a pure tone hearing screening, all participants had monaural thresholds at or below 20 dB HL at all octave frequencies from 250 Hz to 8 kHz in both ears. All participants provided written informed consent, and all procedures were approved by the Institutional Review Board at the University of Washington.

### Instrumentation

Inside a sound-attenuated room, participants were seated approximately 1.5 m from a 23-inch monitor. Their speech was captured by a microphone (WL185, Shure Incorporated, Niles, IL) placed 15 cm from the mouth and connected to an audio interface (RME Babyface Pro, RME, Haimhausen, Germany). The audio interface was connected to a computer with custom software written in MATLAB (The MathWorks, Natick, MA, United States) that recorded the speech signal to computer hard disk. The output of the audio interface was amplified (HeadAmp6 Pro, ART ProAudio, Niagara Falls, NY) and played back to the participant via insert earphones (ER-1, Etymotic Research Inc., Grove Village, IL), providing speech auditory feedback throughout the whole experiment. In addition, the insert earphones were also used to deliver the binaural auditory stimuli for formant discrimination testing and playback of the participant's previously recorded speech in the Listening condition (see below).

Before each recording session, the settings on the audio interface and the headphones amplifier were adjusted such that speech input with an intensity of 75 dB SPL at the microphone resulted in 73 dB SPL output in the earphones [44]. To calibrate the intensity of the speech signal in the earphones, a 2 cc coupler (Type 4946, Bruel & Kjaer Inc., Norcross, GA) was connected to a sound level meter (Type 2250A Hand Held Analyzer with Type 4947 ½″ Pressure Field Microphone, Bruel & Kjaer Inc., Norcross, GA).

### Procedure

The experiment consisted of two parts, a pre-test to record the participant's productions to be used for the creation of the auditory stimuli for formant discrimination testing, and a series of

speaking, listening, and silent reading tasks during which formant discrimination was tested. The pre-test consisted of thirty trials of a speech production task. During each trial, the word "tuck" appeared in green color on a black background and remained visible for 1500 ms. The participant spoke the word when it appeared. After the pre-test was completed, the experimenter used a custom MATLAB script to examine the thirty productions of "tuck" offline and manually mark the onset and offset of the vowel /uh/ (International Phonetic Alphabet symbol /ʌ/) for each trial by visually inspecting the waveform and a wide-band spectrogram. The MATLAB script then extracted the frequencies of the first two formants (F1 and F2) of the middle 20% of each production (a window from 40% to 60% into the vowel duration) as tracked by the Audapter software using the linear predictive coding algorithm [45, 46]. The median F1 and F2 frequencies of the thirty trials were calculated and the pre-test token closest to the median F1 and F2 was selected based on Euclidean distance in the F1-F2 space. The middle 60 ms of the vowel in the selected token was then used to generate the stimuli for formant discrimination testing. Truncated vowels were used so that the two auditory stimuli could be presented back-to-back as close as possible to the time point where PSAM had been demonstrated in our previous studies. Each participant's chosen production was first modified with a linear amplitude envelope to create a 10 ms onset rise and 10 ms offset fall. Next, eleven formant-shifted versions of this truncated vowel were created with the Audapter software by shifting both F1 and F2 upward from 0 to 250 cents in 25 cents increments (i.e., 0 cents, +25 cents, +50 cents, etc.; note that 1200 cents = 12 semitones = 1 octave). To control for any unwanted effects caused by processing in the Audapter software, the processed version with 0-cent shift was used as the standard syllable in the formant discrimination test instead of the original truncated syllable.

The main part of the study included three conditions (Speaking, Listening, and Reading) with two different word lists. Thus, each participant completed six tasks. The order of the conditions and the word lists were randomized for each participant, but within the same word list, the Listening condition always had to be completed after the Speaking condition as the participant's own recorded speech had to be played back in the Listening condition.

Each task consisted of 110 trials. Each trial began with a white word appearing on a black background on a computer monitor (Fig 1). The word was chosen randomly from the applicable word list. The white word remained on the screen for 600 ms. After 600 ms, the color of the word on the screen changed from white to green, and this change in color served as the *go* signal in the Speaking condition. The green word stayed on the screen for 1400 ms. While the word in white characters was displayed on the monitor, the standard stimulus (0 cents shift, 60 ms duration) was first played through the earphones at 290 ms. Then, 100 ms after the end of the standard stimulus (450 ms after the white word appeared), the comparison stimulus was played. The comparison stimulus was randomly selected from the eleven shifted versions of the truncated syllable (0 to +250 cents). The two stimuli were played at ~75 dB SPL (the formant shifting technique sometimes induces a small intensity difference up to ~2 dB). The timing of the two syllable stimuli was chosen such that the pair was centered around the time point for which PSAM had been documented in our previous studies (i.e., 400 ms after presentation onset of the word in white characters and 200 ms prior to the *go* signal; [32–35, 37]).

When the green word disappeared, a prompt "Same Different" was presented on the monitor for the participant to judge whether the standard and comparison stimuli sounded the same or different by pressing either the F key with their left index finger or the J key with their right index finger, respectively, on a keyboard placed on their lap. The prompt disappeared after 1500 ms or as soon as one of the two buttons was pressed. The screen then remained blank for 1000 ms until the next trial started.

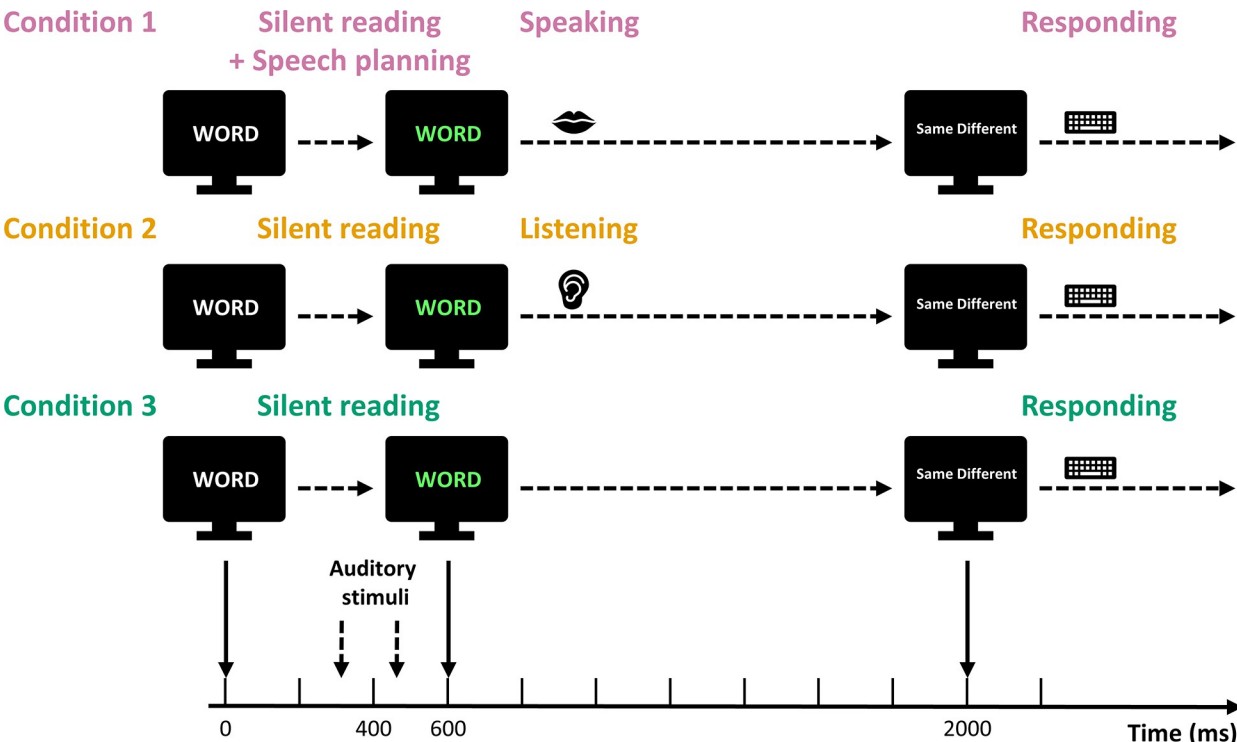

**Fig 1. Experimental procedure.** Each trial started with a white word on a black screen. During the white word period, two truncated vowels, a standard stimulus (at 290 ms) and a comparison stimulus (at 450 ms), were played to the participants. The word changed to green at 600 ms and this color change served as the go signal in the Speaking condition. In the Listening condition, participants listened to playback of their own production after the word changed to green. In the Reading condition, participants silently read the word. At 2000 ms, the green word disappeared and the participants were asked to judge whether the standard and comparison stimuli sounded the same or different by pressing keys on a keyboard.

In the Speaking condition, participants were instructed to say the word on the monitor out loud after the word turned from white to green. In the Listening condition, participants listened to playback of their own production of each word shown on the monitor as recorded during a preceding Speaking condition with the same word list (albeit in different randomized order). Each word was played back with the same intensity and production latency as when it had been actively produced. In the Reading condition, participants were instructed to silently read the words on the monitor without making any articulatory movements.

Each of the two word lists contained 55 CVC words containing three to four letters. To test for a potential effect of congruency between the formant discrimination stimuli and the produced words, one word list ("word list with /uh/") included only words that had /uh/ as their syllable nucleus (e.g., "love", "run") whereas the other word list ("word list without /uh/") excluded any words with /uh/ (e.g., "talk", "sit"). The two word lists were balanced in terms of word frequency [47] and word length.

## Data analysis

For each participant, a logistic regression was fitted to the formant discrimination response data from each of the six tasks using the *glm()* function in the R software [48]. Two parameters, the just-noticeable difference (JND, defined as the shift amount at which the logistic fit predicts a 50% chance of responding "Different") and the slope of the logistic curve were calculated from each fit. The key-pressing response time for each trial was also extracted. Two steps

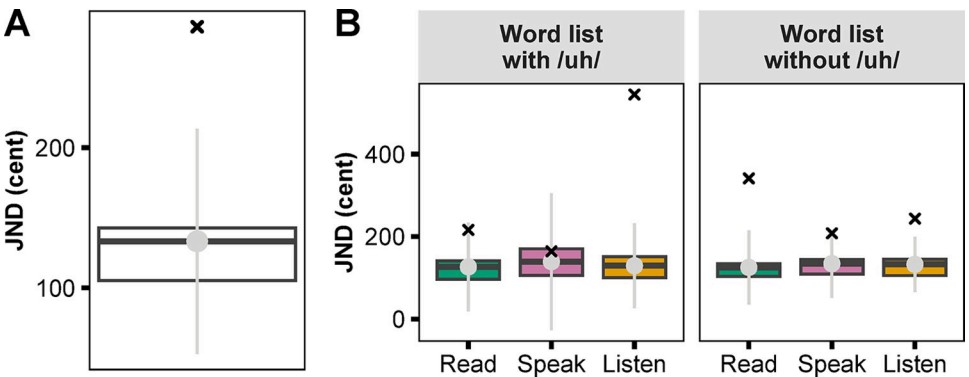

**Fig 2. Boxplots illustrating exclusion of an outlier participant.** (A) just-noticeable difference (JND) averaged across all six tasks. (B) JND by condition (Reading, Speaking, Listening) and word list. The cross symbol (×) indicates the participant who was excluded because the JND averaged across the six tasks was more than three absolute deviations (grey bars) away from the sample median (grey dots).

were taken to exclude data points that were outliers. First, three participants with a negative JND or slope were excluded. Second, the sample distributions of the JND averaged across the six tasks (Fig 2A) and the JND of each task (Fig 2B) were examined. One additional participant was excluded because their JND averaged across the six tasks was more than three absolute deviations away from the sample median [49]. All data from the remaining 22 participants were included in the statistical analyses.

All statistical analyses were conducted in the R software [48]. JND, slope, and response time were used as dependent variables for which we conducted a two-way repeated measures analysis of variance (rANOVA) with Condition (Reading, Speaking, and Listening), Word list ("word list with /uh/" and "word list without /uh/") and their interaction as within-subjects variables. To account for potential violations of the sphericity assumption, the degrees of freedom for within-subject effects were adjusted using the Huynh–Feldt correction [50]. Post-hoc tests of simple effects were conducted by means of paired $t$-tests adjusted with the Holm-Bonferroni method [51]. For effect size calculations, generalized eta-squared ($\eta_G^2$) was used for rANOVA [52] and Cohen's $d$ was used for pair-wise post-hoc tests [53]. Additionally, because the formant discrimination test was a novel test for the participants, we explored potential practice effects with one-way rANOVAs for JND, slope, and response time with Task order (1 to 6) as the within-subject variable, followed by post-hoc $t$-tests. The same adjustment method for multiple comparisons was applied. Lastly, Pearson correlation coefficients were used to examine a potential relationship between response time and either the JND or slope values.

## Results

Fig 3A shows logistic curves fitted to group averaged data for the proportion of "Different" responses at each formant shift level of the comparison stimulus in the six tasks (three conditions by two word lists). For each task, the group averaged JND and corresponding individual participant data are shown in Fig 3B. A two-way rANOVA (Conditions × Word lists) revealed that there was a statistically significant effect for Condition ($F(2.20, 46.12) = 4.82$, $p = 0.01$, $\eta_G^2 = 0.02$), but not for Word list or the interactions. Post-hoc analyses of the Condition effect revealed that the JND in the Speaking condition ($M = 130.77$ cents, $SD = 30.29$ cents) was statistically significantly larger than in the Reading condition ($M = 117.32$ cents, $SD = 30.81$ cents; $t(21) = 3.23$, $p = 0.01$, $d = 0.69$). There was no statistically significant difference in JND between the Speaking and Listening conditions ($M = 123.91$ cents, $SD = 37.59$ cents; $t(21) =$

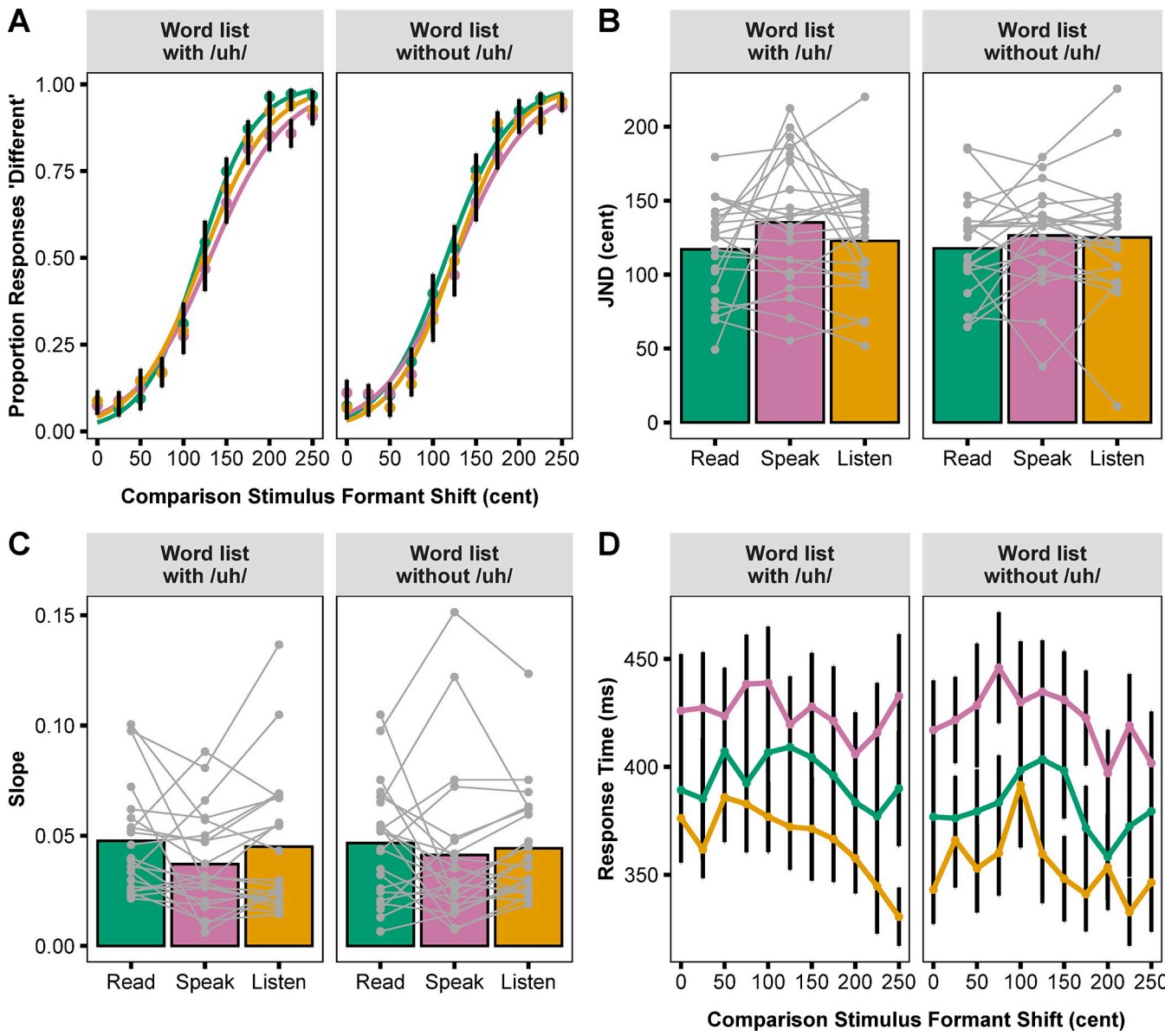

**Fig 3. Formant discrimination task results.** (A) Logistic curves fitted to group averaged data for the proportion of "Different" responses by condition and word list. (B) Mean and individual participant JNDs by condition and word list. (C) Mean and individual participant slopes by condition and word list. JND and slope were calculated from the logistic curves fitted to each participant's responses in each task. (D) Mean keypress response time at each formant shift level of the comparison stimulus by condition and word list. Error bars correspond to standard errors.

1.56, $p = 0.27$, $d = 0.33$) or between the Reading and Listening conditions ($t(21) = -1.49$, $p = 0.27$, $d = -0.32$). A one-way rANOVA with Task order as the within-subjects effect found no statistically significant change in JND with practice ($F(3.61, 75.82) = 1.91$, $p = 0.12$, $\eta_G^2 = 0.03$).

For slope of the fitted curves, none of the main effects or interactions were found to be statistically significant (Fig 3C). However, there was a significant practice effect for slope as revealed by a Task order effect in a one-way rANOVA ($F(4.25, 89.35) = 1.91$, $p < 0.01$, $\eta_G^2 = 0.06$) and post-hoc analyses showing that slope in the first task was statistically significantly smaller than that in the fifth task ($t(21) = -3.39$, $p = 0.04$, $d = -0.72$) and the sixth task ($t(21) = -3.38$, $p = 0.04$, $d = -0.72$).

Fig 3D shows group averaged data for response time at each formant shift level of the comparison stimulus. A two-way rANOVA conducted with the response time data revealed a statistically significant main effect of Condition ($F(1.47, 30.86) = 10.32$, $p < 0.01$, $\eta_G^2 = 0.07$). Post-hoc analyses then revealed that response time in the Speaking condition ($M = 423.88$ ms, $SD = 98.37$ ms) was significantly slower than in the Listening condition ($M = 360.22$ ms, $SD = 67.82$ ms; $t(21) = 7.39$, $p < 0.01$, $d = 1.58$), but there was no significant difference between the Speaking and Reading conditions ($M = 388.08$ ms, $SD = 87.73$ ms; $t(21) = 2.09$, $p = 0.97$, $d = 0.45$) or between the Reading and Listening conditions ($t(21) = 1.86$, $p = 0.97$, $d = 0.40$). Additionally, a one-way rANOVA examining the influence of Task order revealed a statistically significant effect ($F(2.89, 60.62) = 18.32$, $p < 0.01$, $\eta_G^2 = 0.20$). Post-hoc analyses showed a significantly slower response time for the first and second task versus the third, fourth, fifth, and sixth task, and for the third, fourth and fifth task versus the sixth task ($p < 0.05$ in all pairwise comparisons).

Lastly, we calculated Pearson correlation coefficients for the relationship between response time and JND or slope in all three conditions. No statistically significant correlations were found between response time and JND. For slope, there was a significant negative correlation between response time and slope only in the Speaking condition ($r = -0.56$, $p < 0.01$; $p > 0.07$ for all other correlations).

## Discussion

Building upon previous findings of modulated AEPs and weakened intensity discrimination of pure tone stimuli during the speech planning phase, the current study examined whether speech planning modulates speakers' ability to detect small formant frequency differences in recordings of their own vowel productions. The premise was that formant discrimination is critical for auditory feedback monitoring during speech, and that the previously documented phenomenon of PSAM [32–35, 37] may reflect either suppression or selective tuning of auditory neuronal populations in preparation for this feedback monitoring.

Participants performed same/different formant judgments for recordings of self-produced vowels during the speech planning phase before speaking (Speaking condition) as well as prior to passive listening (Listening condition) and during silent reading (Reading condition). We also examined whether congruency between the formant discrimination stimuli and the planned production would affect participants' judgments. Logistic regression functions were fitted to the participants' responses in each condition for both incongruent and congruent word lists. JND was calculated as a measure of formant discrimination sensitivity. We found that participants showed a small but statistically significant decrease in formant discrimination sensitivity (i.e., higher JND) during the speech planning phase in the Speaking condition as compared with the Reading condition. Although other pair-wise comparisons showed no statistically significant differences, the group average JND for the Listening condition fell in-between those for the Speaking condition and the Reading condition. Descriptively, this ranking of JND across the conditions was more clear when the vowels presented for discrimination were congruent with the vowels to be produced, but the influence of congruency was not statistically significant (no main effect or interaction).

In addition to JND, we also determined the slope of the fitted logistic curves as another psychometric measure. Statistical tests showed no significant effects of either independent variable (Condition and Word list) on these slope measures. However, unlike JND, slope showed significant changes over time and it increased from earlier to later tasks. In other words, for slope, there was a practice effect. The interpretation of slope in a same/different discrimination paradigm is not entirely straightforward but relates to the "decisiveness" of a participant's

responses given that a greater slope value indicates a more abrupt transition from "same" responses to "different" responses regardless of the JND value. Keypress response times also exhibited a significant practice effect and decreased from earlier to later trials. Thus, our slope and response time measures indicate that participants' formant discrimination responses became more decisive and faster throughout the experiment, but these behavioral changes did not affect the JND data.

Overall, our results are mostly consistent with those of the only other study that already investigated perceptual correlates of PSAM. Merrikhi and colleagues [36] found that speech planning led to higher discrimination thresholds and higher perceptual uncertainty in a *pure tone intensity* discrimination test. Nevertheless, there are some differences between the results of the two studies. In the current study, the small decrease in formant discrimination sensitivity was only statistically significant for Speaking versus Reading, but not for Speaking versus Listening and also not for Listening versus Reading. On the other hand, in [36], the Speaking condition showed a significantly higher discrimination threshold than both the Reading and Listening conditions. Additionally, we found no effect of condition or word list on the slope of the logistic regression functions as a measure of perceptual uncertainty, whereas [36] found that perceptual uncertainty in their Speaking condition was significantly higher than in their Reading condition.

What may account for these discrepancies between the results from the two studies? First, in both studies the changes in discrimination ability observed during speech planning are very small, and, thus, participant sampling and inter-individual variability may cause inconsistency in terms of whether or not these effects reach statistical significance in a given study. Second, there is prior evidence that speech planning has different effects on the auditory processing of pure tones versus truncated syllables: although the modulation of N1 amplitude seems equivalent for the two types of stimuli, modulation of P2 amplitude, reflecting later stages of auditory processing, may be specific to speech stimuli [32]. Sensory prediction of the speech auditory input in our Listening condition may have a small modulating effect on formant discrimination, thereby reducing the difference in discrimination ability between the Speaking and Listening conditions. Third, the timing of the stimuli for comparison differed between the two studies. In [36], two 50 ms pure tones were separated by 900 ms. In the current study, two 60 ms truncated syllables were separated by only 100 ms such that both tokens could be presented as close as possible to the time point for which PSAM has been documented. It is possible that our paradigm with such a short interstimulus interval made formant discrimination overall more difficult or more variable.

Taken together, the slightly decreased formant discrimination sensitivity during the speech planning phase as compared with during silent reading and the lack of a word list congruency effect are largely consistent with a general auditory attenuation account of PSAM. The difference in formant discrimination between our speaking and listening conditions was not statistically significant, but this result is in keeping with one of our prior EEG studies demonstrating that modulation of the auditory N1 component was also observed before both speaking and listening to prerecorded versions of one's own speech [33]. Analogous auditory predictions may be generated when planning to speak and when expecting to hear, with predictable timing, playback of the same words. Similarly, the absence of a word list effect is not entirely surprising as our previous EEG studies on PSAM have demonstrated that the effect occurs even for a pure tone (a stimulus that clearly lacks congruency with any auditory predictions generated during speech planning) and that the effect PSAM of the N1 component does not differ when the probe stimuli are pure tones versus speech syllables [32].

These results then suggest that a speaking-induced general attenuation of the auditory system already starts during the speech planning phase prior to movement onset, regardless of the acoustic similarity between the auditory input and the predicted acoustic outcomes of the

planned speech movements. Evidence from several other lines of human and nonhuman vocalization studies indicates that, during the actual production, some of the suppressed auditory neurons then respond more strongly when a mismatch is detected between perceived and predicted feedback [6, 13, 54–60].

Nevertheless, a number of alternative interpretations cannot be ruled out at this time. For example, it has been argued that tasks requiring discrimination of same or different syllable pairs recruit sensorimotor networks that are also involved in speech production [61, 62]. This raises the possibility that the requisite activation of these networks during the planning phase in our Speaking condition negatively impacted their contributions to the detection of subtle differences between the discrimination stimuli that were presented during the same time window. In fact, as a more narrow version of this hypothesis suggesting "interference" between sensorimotor processing during speech planning and auditory processing, PSAM may reflect neither a purposeful general suppression nor a fine-tuning of auditory cortex to optimize feedback monitoring but an active involvement of auditory neuronal populations in *feedforward* speech planning. This novel hypothesis certainly is testable, most directly with experimental paradigms examining whether individual participant PSAM measures relate more closely to aspects of speech that reflect the extent of feedforward preparation or, alternatively, that reflect the implementation of feedback-based corrections.

In sum, the current study examined perceptual correlates of PSAM by investigating participants' formant discrimination ability prior to speaking, prior to passive listening, and during silent reading. We found that speech planning led to a small but statistically significant decrease in formant discrimination sensitivity in the absence of a statistically significant effect of congruency between the discrimination stimuli and the predicted acoustic outcomes of the planned speech movements. This work provides new behavioral evidence regarding modulation of the auditory system during speech movement planning and motivates further research into the phenomenon's functional relevance.

## Author Contributions

**Conceptualization:** Hantao Wang, Ludo Max.

**Data curation:** Hantao Wang, Yusuf Ali.

**Formal analysis:** Hantao Wang.

**Funding acquisition:** Ludo Max.

**Investigation:** Yusuf Ali.

**Methodology:** Hantao Wang, Ludo Max.

**Project administration:** Ludo Max.

**Software:** Hantao Wang.

**Supervision:** Ludo Max.

**Visualization:** Hantao Wang.

**Writing – original draft:** Hantao Wang, Ludo Max.

**Writing – review & editing:** Hantao Wang, Yusuf Ali, Ludo Max.

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
