## [Decision Letter · Decision Letter 0]

13 Dec 2023

PONE-D-23-34985Perceptual formant discrimination during speech movement planningPLOS ONE

Dear Dr. Max,

Thank you for submitting your manuscript to PLOS ONE. After careful consideration, we feel that it has merit but does not fully meet PLOS ONE’s publication criteria as it currently stands. Therefore, we invite you to submit a revised version of the manuscript that addresses the points raised during the review process.

Specifically, both reviewers seek further methodological clarifications and justifications related to number of trials and participants which I urge you to provide. In addition, Reviewer 1 makes suggestions for further analyses related to variability and practice effects. Please consider carefully to what extent they can advance your primary research aims and incorporate them accordingly. There are also various minor issues to address as outlined in the reviews.

We look forward to receiving your revised manuscript.

Kind regards,

Vera Kempe

Academic Editor

PLOS ONE

Journal Requirements:

'This research was supported by grants R01 DC017444 and R01 DC020162 (to author L.M.) and T32 DC005361 from the National Institute on Deafness and Other Communication Disorders (https://www.nidcd.nih.gov/). The content is solely the responsibility of the authors and does not necessarily represent the ofﬁcial views of the National Institute on Deafness and Other Communication Disorders or the National Institutes of Health. "

Reviewers' comments:

Reviewer's Responses to Questions

**Comments to the Author**

1. Is the manuscript technically sound, and do the data support the conclusions?

Reviewer #1: Yes

Reviewer #2: Yes

2. Has the statistical analysis been performed appropriately and rigorously? 

Reviewer #1: Yes

Reviewer #2: Yes

3. Have the authors made all data underlying the findings in their manuscript fully available?

Reviewer #1: Yes

Reviewer #2: Yes

4. Is the manuscript presented in an intelligible fashion and written in standard English?

Reviewer #1: Yes

Reviewer #2: Yes

5. Review Comments to the Author

Reviewer #1: In this study, Wang et al. explored the impact of speech planning on speakers' perceptual sensitivity concerning vowel formant discrimination. During the task, participants were tasked with making judgments on pairs of auditory stimuli, consisting of a pre-recorded, self-produced vowel and a formant-shifted version of the same production. The findings revealed that, during the process of speech planning, subjects exhibited reduced perceptual sensitivity in the discrimination of formants, as opposed to conditions where no motor plan was engaged.

The manuscript demonstrates a commendable level of writing, and the analysis appears to be executed with precision. I have only a couple of inquiries, and I look forward to hearing the authors' responses.

I recommend that the authors conduct additional analyses as outlined below:

Given the framework of predictive motor control, where increased focus of the brain on the predictive process may lead to improved estimation of the sensory consequence of motor commands, thereby leading to more accurate motor outputs, it would be valuable to investigate whether there is a correlation between subjects' speech variability and perceptual sensitivity. To explore this, I suggest examining the relationship between the variability of speech output for each subject and their perceptual sensitivity. The variability of speech output can be quantified by calculating the Euclidean distance of each speech output (i.e., trial) from the centroid of the speech outputs in the F1-F2 space.

We can pose the following questions:

1. Relationship between Overall Speech Variability and JND/Reaction Time/Slope:

Investigate if there is a correlation between subjects’ overall speech variability and their JND, reaction time, and slope. For instance, do subjects with higher speech variability tend to exhibit higher JNDs, and vice versa? Additionally, explore the potential impact of task conditions (Speaking, Listening, and Reading) and word congruency on these relationships.

2. Comparison of More and Less Variable Trials:

Classify trials for each subject into two categories based on speech variability in the F1-F2 space: more variable and less variable. Fit a curve for each class and assess whether there are significant differences in subjects’ JND, slope, and reaction times across these trial classes. Investigate the effects of task conditions (Speaking, Listening, and Reading) and word congruency on these differences.

3. Exploring Practice Effects:

Further segregate trials into early (first 55 trials) and late (last 55 trials) categories to examine the effect of practice. Evaluate whether there are significant differences in subjects’ JND, slope, and reaction times between these two trial classes. Explore the potential impact of task conditions (Speaking, Listening, and Reading) and word congruency on these differences.

If any significant relationships are identified during these analyses, I encourage the authors to discuss their relevance in the manuscript. Such discussions can contribute to a deeper understanding of the study's implications and enhance the readers' appreciation of the research.

Minor comment: What do the authors mean by “entire experiment” in line 166: “The entire experiment consisted of 110 trials.” Do they mean each condition-word list combination? Please clarify this.

Reviewer #2: Manuscript: Perceptual formant discrimination during speech movement planning

Decision: Major revisions

This study examines the relationship between speech planning and auditory acuity. There has been previous work done on this, but specifically regarding loudness, which is potentially less relevant for speech motor control. The authors investigate formant acuity, which is potentially an aspect of speech motor control that is being actively predicted and planned during the planning period. The experiment tests acuity during speaking, listening, and reading conditions; in each condition, the vowel tested is either the same as what is produced (e.g. caret vowel tested, “mud” produced) or different (e.g. caret vowel tested, “mead” produced). Auditory stimuli are presented at a time that has previously been shown to be Participants had lower acuity in the speak condition compared to read, but not compared to listening; there was also no significant difference between listen and read. Furthermore, there was no effect of vowel. The authors conclude that this supports a general auditory suppression interpretation of PSAM, rather than an interpretation where modulation is driven by specific tuning of neural populations relevant to the motor task at hand.

Overall this is a well-thought out, well-written study, though with potentially a few caveats depending on further information. There are a few major points that I think should be addressed:

1. Clarity regarding methods. The authors state (L 166) “The entire experiment consisted of 110 trials”. In L 136 the authors refer to the “experiment” as having a pre-test and a series of speaking, listening, and silent reading tasks. So are the authors saying that there were only 110 trials for all 6 conditions? Or were there 110 trials in each condition, for a total of 660 trials? Was each wordlist read once with no repetitions? How many times was each vowel step heard under each condition*word list combination?

2. Power. The Merrikhi et al. 2018 paper only had 16 participants, and this one has 22 that were actually included. I would expect, and the authors seem to agree, that the effects being tested here are likely fairly small, especially the interaction with vowel quality of the spoken word. How did the authors arrive at 26 participants (with data loss to 22) as an appropriate sample size for this effect? In particular, the interpretation for PSAM as a general auditory modulation vs. specific modulation by neural population for the task hinges, as far as I can tell, on two differences that were not demonstrated here: 1. Statistically significant difference between listening and speaking; 2. Significant effect of word list.

Less major points:

3. Alternative interpretation: This interpretation seems like it would predict task-specific effects, such as an effect of word list, which was not found here. If this is incorrect, can the authors provide a little more detail about the neural populations that would be recruited in feedforward planning for speech? It seems that they’d have to be broad—why?

Trivia:

L 37 has an extra space before the period

L 99 don’t understand the use of “already” in this context

L 176 currently “SLP” instead of “SPL”

6. PLOS authors have the option to publish the peer review history of their article (what does this mean?). If published, this will include your full peer review and any attached files.

Reviewer #1: No

Reviewer #2: No

---

## [Author Response · Author response to Decision Letter 0]

1 Feb 2024

See attached document AuthorsResponsesToReviewerComments.docx

---

## [Decision Letter · Decision Letter 1]

19 Mar 2024

Perceptual formant discrimination during speech movement planning

PONE-D-23-34985R1

Dear Dr. Max,

We’re pleased to inform you that your manuscript has been judged scientifically suitable for publication and will be formally accepted for publication once it meets all outstanding technical requirements.

Kind regards,

Li-Hsin Ning

Academic Editor

PLOS ONE

Additional Editor Comments (optional):

Reviewers' comments:

Reviewer's Responses to Questions

**Comments to the Author**

1. If the authors have adequately addressed your comments raised in a previous round of review and you feel that this manuscript is now acceptable for publication, you may indicate that here to bypass the “Comments to the Author” section, enter your conflict of interest statement in the “Confidential to Editor” section, and submit your "Accept" recommendation.

Reviewer #1: All comments have been addressed

Reviewer #2: All comments have been addressed

2. Is the manuscript technically sound, and do the data support the conclusions?

Reviewer #1: Yes

Reviewer #2: Yes

3. Has the statistical analysis been performed appropriately and rigorously? 

Reviewer #1: Yes

Reviewer #2: Yes

4. Have the authors made all data underlying the findings in their manuscript fully available?

Reviewer #1: No

Reviewer #2: Yes

5. Is the manuscript presented in an intelligible fashion and written in standard English?

Reviewer #1: Yes

Reviewer #2: Yes

6. Review Comments to the Author

Reviewer #1: (No Response)

Reviewer #2: (No Response)

7. PLOS authors have the option to publish the peer review history of their article (what does this mean?). If published, this will include your full peer review and any attached files.

Reviewer #1: No

Reviewer #2: No

---

## [Editor Report · Acceptance letter]

22 Mar 2024

PONE-D-23-34985R1 

PLOS ONE

Dear Dr. Max, 

I'm pleased to inform you that your manuscript has been deemed suitable for publication in PLOS ONE. Congratulations! Your manuscript is now being handed over to our production team.

Kind regards, 

on behalf of

Dr. Li-Hsin Ning 

Academic Editor

PLOS ONE